# CroReLU: Cross-Crossing Space-Based Visual Activation Function for Lung Cancer Pathology Image Recognition

**DOI:** 10.3390/cancers14215181

**Published:** 2022-10-22

**Authors:** Yunpeng Liu, Haoran Wang, Kaiwen Song, Mingyang Sun, Yanbin Shao, Songfeng Xue, Liyuan Li, Yuguang Li, Hongqiao Cai, Yan Jiao, Nao Sun, Mingyang Liu, Tianyu Zhang

**Affiliations:** 1Department of Thoracic Surgery, The First Hospital of Jilin University, Changchun 130012, China; 2School of Instrument and Electrical Engineering, Jilin University, Changchun 130012, China; 3Department of Hepatobiliary and Pancreatic Surgery, The First Hospital, Jilin University, 71 Xinmin Street, Changchun 130021, China; 4Center for Reproductive Medicine and Center for Prenatal Diagnosis, The First Hospital of Jilin University, Changchun 130012, China

**Keywords:** pathology image, lung cancer, deep learning, image classification

## Abstract

**Simple Summary:**

In the clinical diagnosis of lung cancer, doctors mainly rely on pathological images to make decisions about the patient’s condition. Therefore, pathology diagnosis is known as the gold standard for disease diagnosis. In recent years, convolutional neural networks have been widely used for computer-aided diagnosis to relieve the work pressure of pathologists. However, this method still faces two problems: from the perspective of clinical application, the current neural network model can only perform simple lung cancer type detection; From the perspective of model design, the strategy adopted by researchers to improve the accuracy of diagnosis is often to carry out complex model design, which will cause the model parameters to be too large to be clinically deployed. In this study, we first prepared a lung cancer dataset that can provide more complex cancer information to the model. Then, using only a novel visual activation function, the ability of convolutional neural networks to detect interclass and intraclass differences in cancer pathological images is enhanced.

**Abstract:**

Lung cancer is one of the most common malignant tumors in human beings. It is highly fatal, as its early symptoms are not obvious. In clinical medicine, physicians rely on the information provided by pathology tests as an important reference for the final diagnosis of many diseases. Therefore, pathology diagnosis is known as the gold standard for disease diagnosis. However, the complexity of the information contained in pathology images and the increase in the number of patients far outpace the number of pathologists, especially for the treatment of lung cancer in less developed countries. To address this problem, we propose a plug-and-play visual activation function (AF), CroReLU, based on a priori knowledge of pathology, which makes it possible to use deep learning models for precision medicine. To the best of our knowledge, this work is the first to optimize deep learning models for pathology image diagnosis from the perspective of AFs. By adopting a unique crossover window design for the activation layer of the neural network, CroReLU is equipped with the ability to model spatial information and capture histological morphological features of lung cancer such as papillary, micropapillary, and tubular alveoli. To test the effectiveness of this design, 776 lung cancer pathology images were collected as experimental data. When CroReLU was inserted into the SeNet network (SeNet_CroReLU), the diagnostic accuracy reached 98.33%, which was significantly better than that of common neural network models at this stage. The generalization ability of the proposed method was validated on the LC25000 dataset with completely different data distribution and recognition tasks in the face of practical clinical needs. The experimental results show that CroReLU has the ability to recognize inter- and intra-class differences in cancer pathology images, and that the recognition accuracy exceeds the extant research work on the complex design of network layers.

## 1. Introduction

According to the global epidemiological statistics on tumors published by the American Cancer Society in 2018, there are about 2.1 million new cases of lung cancer worldwide, accounting for 11.6% of total cancer incidence, and about 1.8 million deaths, accounting for 18.4% of total cancer deaths. Among them, the incidence of lung adenocarcinoma is particularly remarkable, having accounted for more than 40% of non-small cell lung cancer and surpassing squamous cell carcinoma as the most common histological category of lung cancer. Invasive lung adenocarcinoma is the most common among lung adenocarcinomas, and generally 70–90% of surgically resected lung adenocarcinomas are invasive. Due to the lack of obvious clinical symptoms in the early stage, it is often difficult to detect such cases by conventional means of detection. At the same time, the tumor has the phenomenon of being prone to metastasis in the early stage, thus making clinical diagnosis difficult. The accurate and comprehensive detection of the clinicopathological infiltration degree is undoubtedly important to guide the clinical treatment of lung adenocarcinoma.

In current disease diagnosis, the most common diagnostic basis used by physicians comes from relevant images captured by medical devices, as this approach can visually help them analyze the disease. The most reliable basis for this test is the biopsy of tissue cut from the patient’s body to obtain pathological sections. This plays an important role in both clinical medicine and scientific research. However, when using pathology slides, in order to make full use of their information it is necessary to solve the following problems. First, the process of making manual pathology slides is relatively complicated, and diagnosis takes a long time. This requires pathologists with a certain level of experience to carefully search for lung cancer tissues using microscopes with different magnifications. Then, based on relevant biological characteristics such as the heterogeneity of cancer cells, the ratio of tumor parenchyma to mesenchyme, and the ratio of cancer cells, it is possible to determine whether the patient has lung cancer and the extent of the disease, then take effective measures.

At this stage, it is time-consuming and inefficient for pathologists to manually count various biological features, and there is a risk that key features may be missed due to overwork, resulting in misdiagnosis. In addition, the doctor’s review experience and the number of films reviewed have a direct impact on the accuracy and validity of diagnostic results. Therefore, the use of deep learning technology to assist in the diagnosis of whole slide imaging can both greatly reduce the workload of physicians and alleviate the problem of uneven distribution of pathologists, especially in underdeveloped areas, and further reduce the probability of missing diagnoses in the process of pathology diagnosis, which has valuable clinical significance [1,2,3].

With the rapid development of deep learning and computer vision research, algorithms based on deep learning have been applied to various complex intelligent recognition fields such as image, text, and audio, achieving remarkable results. For example, image classification [4,5,6], object detection [7,8], instance segmentation [9,10], natural language processing [11], and speech recognition [12] have coincidentally surpassed normal human capabilities.

The activation function (AF) has the property of learning abstract features by nonlinear transformation, and is designed to help neural networks learn complex patterns in data. AF plays an irreplaceable role in convolutional neural networks [13]. At first, the sigmoid function [14] or tanh function were generally chosen for the AF of neurons; however, in the landmark convolutional neural network AlexNet [15], the nonlinear unsaturated function (ReLU) [16] was chosen for the first time. In terms of gradient decay during training, ReLU only required five epochs of iterations to reach a training error rate of 25% for the deep network; however, the tanh unit needed 35 epochs of iterations, and the training time with ReLU was only one-sixth the time required with tanh. Subsequently, VGG [17] and ResNet [18] have used ReLU as the AF in convolutional networks and have achieved great success.

As a result, although novel AF research work has emerged in the subsequent network model design process, researchers now seem to default to ReLU as the best AF, making their successors tend to adopt the strategy of increasing the depth and width of the network or changing the convolutional network components in the model optimization process for the corresponding tasks.

Xu [19], for example, used an EM-CNN, which was implemented in two stages. The first stage was to locate the identification region, and the second stage was to achieve the classification of lung adenocarcinoma, lung squamous carcinoma, and normal tissues. The AUC of normal and tumor tissue classification was 0.9978, and the AUC of lung adenocarcinoma and lung squamous carcinoma classification was 0.9684. Tsirigos et al. [20] retrained Google’s deep learning algorithm Inception V3 in 2018 using a large number of digitized pathology section images. The AI achieved 99% accuracy in identifying cancerous and normal tissues and 97% accuracy in distinguishing adenocarcinoma and squamous carcinoma. The algorithm was able to identify the six common mutations in lung cancer, including EGFR mutations, for which targeted therapies are available, from the section images with an accuracy of 73–86%. Adu.K et al. [21] proposed a dual horizontal squash capsule network (DHS CapsNet) based on capsule networks for the automatic diagnosis of histopathological images of lung and colon cancers. Its greatest highlight is the use of encoder fusion features to make the obtained information richer followed by the use of a horizontal squash (HSquash) function to effectively compress the vectors, which allows the network to effectively extract important information. A deep neural network framework was developed by Wang et al. [22] for identifying lymph nodes and tumor regions in digital pathology images. It first feeds the case images to be processed into a segmentation network for extracting valid lymph node sites. Then, it uses a classification network to classify cancer in the region, allowing it to be measured, and visualizes the computed tumor sites. The authors eventually concluded that this deep learning model was able to both detect histological features of tumors and help doctors to analyze prognostic conditional factors. Khosravi [23] built a CNN_Smoothie pipeline based on three convolutional neural networks with powerful generalization ability; it was able to effectively distinguish two lung cancer subtypes, four bladder cancer biomarkers, and five breast cancer biomarkers, with recognition accuracies of 100%, 92%, and 95%, respectively. Riasatian et al. [24] proposed a variant of a dense residual network capable of fine-tuning and training on pathology images. They evaluated this approach with three publicly available datasets (TCGA, endometrial cancer images, and colorectal cancer images). The network was able to extract richer information on histopathological image features compared with the original convolutional neural network.

Although these methods achieve high accuracy, they have two common problems: (1) most of the functions are for squamous and adenocarcinoma diagnosis of lung cancer, without further functions such as cancer infiltration level screening, which is more limited in clinical effectiveness; (2) the ReLU AF as most often used is a general AF for deep learning, and this AF is directly applied to the image classification task of pathological sections, which has the following drawbacks:(1)The forced sparse processing of the ReLU AF reduces the effective capacity of the model, and the clearing of negative gradient values at x < 0 may result in neurons that are no longer activated by any data, leading to neuron ‘necrosis’.(2)The ReLU AF is not specifically used for computer vision tasks, no information about adjacent features is noted, and the AF is not spatially sensitive.(3)Most of the pathological features of lung cancer show tubular morphology such as papillae, micropapillae, and apposition, which the ReLU AF may not be able to capture.

However, researchers have seemed to ignore the shortcomings of the AF, instead aiming to find an optimal model for the diagnosis of pathological slice images from the perspective of migration learning or in the structural design of neural networks. This has the expense of introducing cumbersome structures and making the model heavier, which objectively makes clinical deployment more difficult.

**Does a relatively simple method exist to achieve similar results?** In this regard, the present work focuses on the spatial perspective of the AF. By analyzing the biohistological features of lung cancer pathology images, a crossover spatial AF CroReLU is designed based on observed a priori knowledge, converting the ReLU from a scalar input to an input incorporating two-dimensional (2D) spatial information. This enables the convolutional network to capture complex lung cancer pathologies after a simple replacement of the AF morphology. Specifically, this work addresses the following aspects of the deep-learning-based digital pathology slice diagnosis task:(1)A novel AF called CroReLU is designed based on prior knowledge of pathology; it has the ability to model crossed spaces, and can effectively capture histological shape features such as lung cancer blisters, papillae, and micropapillae without changing the model network layer structure.(2)The proposed method uses a plug-and-play visual activation that can be applied to any state-of-the-art computer vision model for image analysis-related tasks.(3)A digital pathology image dataset for lung cancer infiltration level detection was prepared by a pathologist, and the experimental results demonstrate that CroReLU can sensitively capture infiltrative and microinfiltrative features, and possesses the potential to solve practical clinical tasks.

The outline of this paper is as follows: Section 2 details the process of experimental data preparation. Section 3 mainly describes the proposed CroReLU method. Section 4 discusses the performance of CroReLU for pathological image recognition and the reasons for it, further corresponding ablation experiments explaining its effectiveness, and extended experiments to verify the generalization performance of the method. Finally, Section 5 summarizes the work and suggests future directions for exploration.

## 2. Dataset

The pathology digital section image data were obtained from 766 cases in 2019–2020 at the First Hospital of Baiqiu’en, Jilin University, and the data were compiled and labeled by four professionals each with more than five years of work experience over a period of three months to ensure that each sample case had significant clinical phenomena. The selected data were then used to producde digital images using a digital pathology scanning instrument and cropped into patch images at 20× magnification, with each section size being 224 × 224. In addition, patches that were blurred or over-stained as well as full-slide backgrounds were screened out through a manual process during the data pre-processing stage. Finally, as shown in Figure 1, the dataset contained a total of 7332 lung cancer pathology sections that could be used in the experiments, of which 2444 were assigned to the Infiltration category, 2444 were diagnosed as Microinfiltration, and the other 2444 belonged to Normal cases. We randomly selected 870 images (290 Infiltration, 290 Microinfiltration, 290 Normal) as the test set and used the results from the test set feedback to continuously optimize the model parameters to ensure that its performance met the clinical criteria. Among the invasive and microinvasive lung adenocarcinomas, the tumors were predominantly of the adnexal type of growth, whereas microinvasive adenocarcinomas had foci of infiltration ≤0.5 cm in maximum diameter and were of the follicular, papillary, micropapillary, or solid type. Infiltrating adenocarcinomas had foci of infiltration with a minimum diameter >0.5 cm. Normal represented a benign area of lung tissue. The detailed production process of this experimental dataset is shown in Figure 2.

## 3. Neural Network Model

In this experiment, the SE-ResNet-50 network was used. SENet can automatically obtain the importance of each feature channel, then boost the useful features and suppress the features that are not useful for the current task according to this importance. In this experiment, the AF ReLU was replaced by CroReLU, which was designed specifically for pathology slices.

### 3.1. ReLU

ReLU is widely used for computer vision tasks because it requires only one max(·) operation, which is computationally cheaper compared with the exponential operations of tanh and sigmoid. ReLU can accelerate learning and simplify models by increasing their sparsity. The formula for ReLU is shown in Figure 3b. The value of *f*(*x*) is 0 when *x* ≤ 0 and the value of *f*(*x*) is *x* when *x* > 0.

In fact, ReLU has the obvious problem that if the gradient of the function of the input is too large, it can cause the network weights to be updated too much at once. Furthermore, it is possible that for any training sample $x_{i}$ the output of the neural network can be less than zero. The fact that ReLU has a constant zero derivative in the zone where the input is negative makes it particularly sensitive to outliers. This may cause it to shut down permanently, which results in neuron death and gradient disappearance.

### 3.2. CroReLU

During data analysis, we found that the histological features of lung cancer pathology can be broadly classified into adnexal, alveolar, papillary, micropapillary, and solid types, and the cells that are likely to be presented in the high magnification field of view are mostly tubular, columnar-elliptical, and peg-shaped. Pathologists rely on the information of these complex shape features in order to diagnose the type of cancer and the degree of infiltration. From the visual sensory point of view, the images are very neighborly (adjacent pixels are likely to be the same), which requires the neural network to be more sensitive to spatial information. In general, if the goal is to enhance the model’s ability to spatially model the aforementioned tissue shapes, it is necessary to design more complex convolutional layers to provide the network with this ability, ignoring the consideration of the spatial dimension of the activation layer. As shown in Figure 3a, we first analyzed the approximate location of the pathological feature shapes on the feature map on the lung cancer pathology image; it can be seen that each pixel point in the output feature map represents a block in the original image due to the CNN downsampling operation. Assuming that a single tubular lesion is represented by three scalars (red dots) in the vertical direction on the feature map, as in the figure, the ReLU function behaves as in Figure 3b, where only one scalar can be computed during each activation, each scalar contains only its own part of the information, and it does not have the ability to model the contextual information. To address this problem, we propose a new activation function, CroReLU, as shown in Figure 3c, which utilizes a crossed receptive field for spatial context-based activation of each scalar, replacing the original ReLU function. This unconventional square perceptual field window fully considers the histological shape characteristics of lung cancer. Two 1D convolutions, *K*∗1 and 1∗*K*, are utilized to equate to a 2D cross-shaped window for spatial context modeling. In this way, each scalar of the feature map contains information about the shape of pegs, papillae, etc. Regarding the design method of the cross-shaped window, it was proven in [25] that by assuming several size-compatible 2D convolution kernels while keeping the step size constant on the same input, they should have the following characteristics:(1)$$I*\left(K^{\left(1\right)}{+K}^{\left(2\right)}\right)=I*K^{\left(1\right)}+I*K^{\left(2\right)}\;\;\;\;\;\;\;$$

Here, we denotes the input matrix and $K^{\left(1\right)}$, $K^{\left(2\right)}$ represent the size-compatible 2D convolution kernels. CroReLU follows the design idea of ReLU by using a simple nonlinear max(·) to obtain the maximum value between the input x and the condition. Instead, for the core condition design, we extend the theory of 2D convolutional kernel additivity to the activation layer, making the AF dependent on the spatial context in the feature map, which contrasts with recent design approaches where the condition is referenced to the features of the pathological histological shape in the context. We define this formally as *L*(*x*). As in Figure 3c, we define it specifically as follows:(2)$$f\left(x_{c,i,j}\right)=\max\left(x_{c,i,j},L\left(x_{c,i,j}\right)\right)\;\;\;\;$$
(3)$$L\left(x_{c,i,j}\right)=\;x_{c,i,j}\cdot P_{c}^{w}={x\;}_{c,i,j}\cdot P_{c}^{w1}+{x\;}_{c,i,j}\cdot P_{c}^{w2}$$

Here, *f*(·) denotes the nonlinear AF, $x_{c,i,j}$ represents the nonlinear activation input of the *C*th channel on the 2D space (*i*,*j*), the function *L*(·) represents the crossover condition, $P_{c}^{w1}$ represents the convolution kernel operation, with ${x\;}_{c,i,j}$ as the center kernel size of $K_{h}\times 1$. In addition, $P_{c}^{w2}$ represents the convolution kernel operation with a kernel size of $1\times K_{w}$ under the same input, $P_{c}^{w}=P_{c}^{w1}+P_{c}^{w2}$, which represents the 2D AF on $x_{c,i,j}$ on the 2D space for cross-fusion, ensuring that more perceptual field information is obtained with as little computation as possible.

CroReLU as a stand-alone module can easily replace the normal AF. However, it needs to be combined with the convolutional and BN layers, which are the neural network components, for the activation layer to work. According to [26], the components of a neural network are “pre-activation” (BN -> ReLU -> Conv) and “post-activation” (Conv -> BN -> ReLU). For image classification tasks, using “pre-activation” has two advantages: (1) using BN as pre-activation enhances the regularization of the model; and (2) it makes the network easier to optimize. To explore the performance and plug-and-play nature of CroReLU, we summarize two neural network components based on a previous work: Conv-ReLU (“post-activation”) and CroReLU-Conv (“pre-cross-activation”). As shown in the Figure 4a, the network components that constitute Conv-ReLU are Conv -> BN -> ReLU, while the composition of CroReLU-Conv is BN -> CroReLU -> Conv (Figure 4b). In the process of building the neural network model, as shown in Figure 4c, we selected SeNet as the backbone and inserted CroReLU-Conv directly to replace the original network components. Because SeNet can focus on the relationship between channels, it is expected that the model can automatically learn the importance of different channel features. For this purpose, SeNet uses the Squeeze-and-Excitation (SE) module, which aims to model the correlation between different channels; by learning, the network automatically obtains the important features of each channel. Finally, different weight coefficients were assigned to each channel to reinforce the essential features and suppress the unimportant features.

## 4. Experiment

### 4.1. Data Augmention and Experimental Setup

To be able to flexibly cope with the complexity in clinical processing, we used data enhancement techniques to enhance the heterogeneity of the relevant images in the lung cancer pathology training set. Data augmentation can increase the number and diversity of pathology samples in the training set, improve the stability of model performance with noisy data, and increase the generalization ability of the neural network. In the training mode, we first data-enhanced the input images of the training set using random horizontal flip, vertical flip, random rotation, and random noise overlay operations. Then, we manually checked each enhanced image to ensure that each slice positively contained valid regions. In this way, it could be ensured that the pathological regions of the sample were all subject to a small amount of random perturbation. For each image, this was performed three times randomly using the appeal method. Finally, the enhanced training samples were uniformly scaled to a size of 224 × 224 for the enhanced training samples.

For the hyperparameter settings, the rectified Adam (RAdam) optimizer was chosen to optimize the network parameters with a batch size of 64. The initial learning rate was set to 0.0001 (10 decays per 20 steps), and the neural network model used in the experiments ensured that the training was completed within 200 epochs. The models evaluated were implemented using the PyTorch 1.8.0 framework with NVIDIA Cuda v8.0 and cuDNN v10.1 accelerated libraries, and were coded using Python 3.7. All experiments were conducted using Windows 10 on a machine with an Intel Core i9-10875H 2.30 GHz CPU, GPU NVIDIA RTX 3090, and 32 GB RAM.

To evaluate the diagnostic performance of the model for cancer pathology, we used the overall accuracy, sensitivity, specificity, and precision as the evaluation metrics and compared the results of the proposed method with those of the state-of-the-art model. The overall accuracy indicates the proportion of samples correctly predicted by the model to the total number of samples, the precision indicates the proportion of samples correctly predicted by the model to the total number of positive samples, the sensitivity indicates the proportion of samples correctly predicted by the model to the total number of positive samples, and the specificity indicates the proportion of samples correctly predicted by the model to the total number of negative samples:(4)$$\mathrm{Accuracy}=\frac{TP+TN}{TP+TN+FP+FN}$$
(5)$$\mathrm{Precision}=\frac{TP}{TP+FP}$$
(6)$$\mathrm{Sensitivity}=\frac{TP}{TP+FN}$$
(7)$$\mathrm{Specificity}=\frac{TN}{TN+FP}$$

True positive (*TP*): An instance is a positive sample and is predicted to be a positive sample.

False negative (*FN*): An instance is a positive sample and is predicted to be a negative sample.

False positive (*FP*): An instance is a negative sample and is predicted to be a positive sample.

True negative (*TN*): An instance is a negative sample and is predicted to be a negative sample.

### 4.2. Experimental Results on a Private Dataset

For the lung cancer pathology image classification problem, we selected SENet50 and MobileNet with 50-layer depth as the baseline models. SENet is the champion model of the image competition ImageNet 2017, which largely reduces the error rate of previous models and makes the plug-and-play class of model design widely promoted and recognized. MobileNet, a lightweight convolutional neural network focused on mobile or embedded devices, has been widely used since its inception. Our visual activation can be easily applied on top of the network structure by simply replacing the ReLU in the above CNN structure (as shown in Figure 4b). For comparison, we refer to the modified methods as SENet50_CroReLU and MobileNet_CroReLU. All models were trained with the same hyperparameter settings as in Section 4.2, and the diagnostic results are reported in Table 1. Compared with the original SENet, the AF with added spatial information increases the accuracy from 96.32 to 98.33, precision from 96.51 to 98.38, and sensitivity by 2.02. In addition, CroReLU can improve the performance effect of the lightweight model. As can be observed from the table, the accuracy of MobileNet increased from 95.40 to 97.01 after the replacement of the AF, and the precision and sensitivity increased from 95.46 and 95.42 to 97.07 and 97.04, respectively. These results demonstrate that as long as the histological shape features of lung cancer are fully taken into account at the activation level, the effectiveness of pathological image classification can be improved even without complex structural design of the network.

The experimental results can be further addressed by the relevant model confusion matrix shown in Figure 5. As can be seen from Figure 5a,c, the errors in the classification process of SENet50 and MobileNet mainly focus on the confusion between Induction and Micro Infitration, as the samples of both categories of appeal contain tumors and the difference only lies in whether the infiltration area is greater than 5 mm. For certain samples near the critical value, the model provides an incorrect judgment; intuitively, this is consistent with the visual misconception of human observation of features. Conversely, as can be seen in Figure 5b, SENet50_CroReLU effectively mitigates this problem, with only one case of an image being confused between the Infiltration and Microinfiltration categories. As can be seen from Figure 5d, MobileNet_CroReLU is not effective in solving the obfuscation problem, although it has improved performance. We analyze the reasons for this in two ways. First, MobileNet [27] uses deep separable convolution instead of traditional convolution, which sacrifices 1% of the accuracy in return for reducing the computation time by ninefold, suggesting that MobileNet itself aims to sacrifice a small amount of accuracy to lighten the convolutional neural network and facilitate the relevant deployment of the model. Second, the biggest difference between MobileNet and SENet is that SENet is designed with a channel attention mechanism (SE module), and it is the incorporation of the SE module that makes SENet steadily outperform ResNet, MobileNet, and other models in various computer vision tasks. Thus, CroReLU used in combination with the SE module can effectively identify differences in infiltrative and microinfiltrative pathological features.

### 4.3. Ablation Experiment

The results in Section 4.2 showed that CroReLU achieved encouraging results in the diagnosis of pathological infiltrative lung cancer due to its good spatial modeling capability. Thus, we verified the effect of each pixel spatial region size on model performance through ablation experiments. While keeping the experimental parameters uniform, only the size of the crossover window was changed to 3 × 3, 5 × 5, 7 × 7. SENet denotes the case of using only ReLU, and “SENet_CroReLU” denotes the case of using CroReLU as the AF and making the spatial regions of the crossover correspond to 3 × 3, 5 × 5, 7 × 7. The comparison results are shown in the Table 2, it show that the 3 × 3 region provides the best diagnostic effect for the model. While the larger cross-space region shows a higher effect than SENet, it does not exceed the benefit of 3 × 3. We compared the effect of CroReLU on the number of model parameters as well. In the model parameters, the cross-space modeling of 3 × 3 increases the number of covariates by only 0.6 M, and even the largest, 7 × 7, increases it by only 1.7 M. In the inference phase, the average test time per pathology image for SENet_3 × 3 is about 0.393 s. For the normal SENet, the average test time per image is about 0.372 s. Even for the SENet_7 × 7 model, the test time only increases by 0.133 s compared with the original model. This indicates that although the time complexity of CroReLU is slightly higher than that of the scalar visual activation method, its test time required meets the range of pathology clinical diagnosis requirements.

In addition, Figure 6 shows the relationship between the accuracy rate and the percentage of CroReLU contained in the model. In this experiment, three experimental models (SENet50, MobileNet, and ResNet50) were selected; their common feature is that they are all composed of four blocks in the structural design. The horizontal coordinate of 0 represents the accuracy obtained when using ReLU in all blocks of the model. Block1 indicates that CroReLU is applied only in the first block of the model, and the rest use ReLU for visual activation. Block2 represents that CroReLU is used in the first two blocks of the model and ReLU is maintained in the last two blocks. Block3 indicates that only the last block uses ReLU, and Block4 indicates that all AFs in the model use CroReLU. As shown in the figure, when the AFs in the first two blocks are replaced (especially when only Block1 is used), the performance of MobileNet and ResNet50 is not greatly improved, which indicates that the receptive field extracted by the convolution network in the middle and low layers is relatively small and the advantage of activation considering pixel space information is not exploited. By contrast, SENet has the advantage of channel attention mechanism, and shows a certain upward trend in accuracy in the early stage. After Block2, the accuracy of MobileNet and ResNet50 is significantly improved, which proves that CroReLU needs feature maps containing sufficient semantic information to perform at a high level. The channel attention mechanism needs to be fully paired with CroReLU to show the best results in pathology diagnosis. In summary, all visual activation methods need to be replaced with CroReLU to take advantage of neural networks in cancer pathology diagnosis.

### 4.4. Extended Experiment

To ensure that SENet50_CroReLU is not sensitive only to the degree of lung cancer infiltration, we validated the generalization ability of the proposed method on the LC25000 dataset [28], which contains pathology images of both lung and colon cancer divided into five major categories: conventional lung cell digital pathology images (Lung_n), lung adenocarcinoma digital pathology images (Lung_aca), lung squamous adenocarcinoma digital pathology images (Lung_scc), colon adenocarcinoma digital pathology images (Colon_n), and normal colon cell digital pathology images (Colon_aca). As can be seen from the Table 3, the number of data samples for each disease category is as high as 5000, and the total number of samples in the dataset is 25,000, which supports the reliability of the experimental results. As with the private lung cancer dataset, each of these categories was divided into a training set (4500) and a test set (500). Examples of samples from each class of the LC25000 are shown in Figure 7.

Figure 8 shows the effect of SENet50_CroReLU on LC25000 using the confusion matrix and the receiver operating characteristic (ROC) curve. To ensure the fairness of the comparison experiment, the training parameters of the model on the public data were the same as those of the private dataset. The confusion matrix and ROC curves show that this method almost accurately identifies the samples to be tested in each category. This result indicates that CroReLU incorporating a priori knowledge can effectively identify intra-class differences between lung cancer and rectal cancer while acutely trapping inter-class differences between them.

As a visual AF specifically designed for lung cancer pathology detection, we responsibly used the frontier methods proposed by Masud M. et al. [29], Mangal S. et al. [30], and B.K. Hatuwal et al. [31] as comparative models and ensured that these methods were used experimentally on LC25000. Specifically, Masud M. et al., proposed 2D-DFT and 2D-DWT pathological feature extraction methods and used them in combination with CNN. Mangal S. et al., sought to diagnose lung cancer and rectal cancer using two different shallow CNNs. However, the problem with this approach is that it is not possible for a model to distinguish between the subtypes of intestinal and lung cancer at the same time, and it can only perform intra-class recognition tasks, not inter-class recognition tasks. By contrast, B.K. Hatuwal et al. used CNN to perform diagnostic studies only on pathological images of lung cancer tissues, and did not take intestinal cancer into account. Table 4 shows the comparison between the proposed method and the methods mentioned above. Without excessive design of the network structure, SENet50_CroReLU only replaces one visual AF, resulting in a model with an overall accuracy level of 99.69 (+1.8) and significantly better precision and sensitivity than similar models (+2.54, +2.53). The experimental results demonstrate that incorporating only spatial information based on an a priori design for visual activation can effectively improve the model’s diagnostic capability for pathological images. Another advantage of this approach over recent work is that it avoids the complexity of network parameters and reduces the difficulty of model deployment in clinical treatment.

## 5. Conclusions and Future Work

We combined a priori knowledge to design a visual AF incorporating cross-crossing spatial information to capture symmetric and asymmetric histological morphological features such as glandular vesicles, papillae, and micropapillae in lung cancer. This may provide other research ideas in the field of intelligent diagnosis of pathology in addition to the design approach of convolutional networks for deep learning models. To further improve the ability of the model to handle pathological features, we introduced a channel attention mechanism to be used with CroReLU. The results show that our approach is significantly more advantageous that alternatives in determining the degree of lung cancer infiltration and diagnosing the cancer subtype, and has the potential to be easily deployed at the margins in the clinic thanks to the avoidance of model overcomplication.

In future studies, we intend to revisit the pathological sections of cancers and prepare more detailed pathological subtype labels, classifying them into categories such as adnexal predominant, glandular vesicle predominant, papillary predominant, and micropapillary predominant to allow the model can learn more precise and detailed knowledge and provide more professional diagnostic references. We intend to expand the experimental scope in order to establish an effective link between digital pathology section images and the prognostic, survival, and genetic characteristics of lung cancer, allowing our method to be used in assessing the prognosis and survival of lung cancer by analyzing pathology images. After obtaining a highly reliable model, we will conduct research on pathology image quality assessment using neural networks, thus lowering the threshold for preparing medical datasets and creating more convenient experimental conditions for our colleagues.

## Figures and Tables

**Figure 1 cancers-14-05181-f001:**
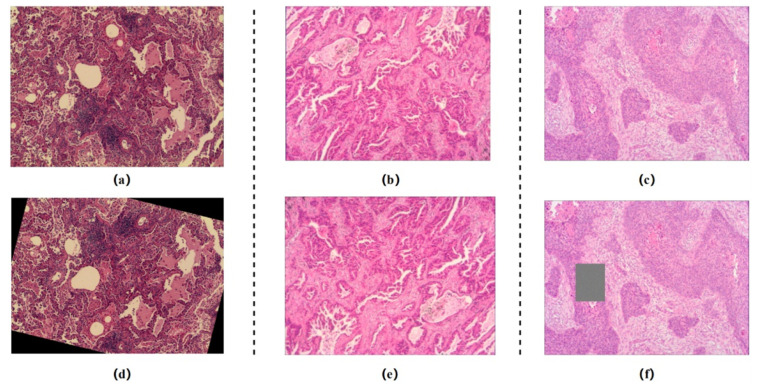
Lung adenocarcinoma data: representative pathological images of (**a**) microinvasive lung adenocarcinoma, (**b**) invasive lung adenocarcinoma, and (**c**) normal lung tissue; data enhancement operations: (**d**) random sample rotation, (**e**) random flip, and (**f**) random region masking. (The initial size of the images are 2048 × 1536 and are resized to 224 × 224).

**Figure 2 cancers-14-05181-f002:**
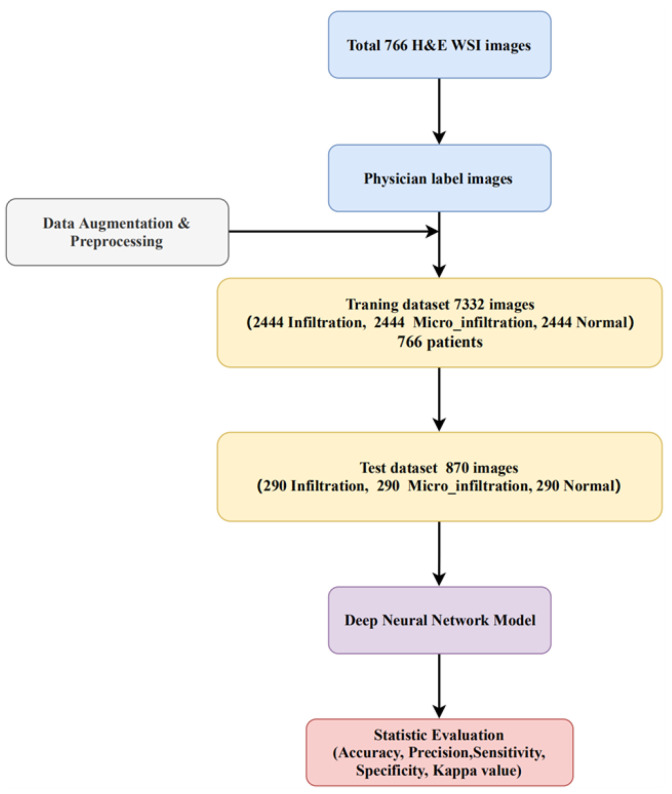
Overall experimental data preparation and workflow.

**Figure 3 cancers-14-05181-f003:**
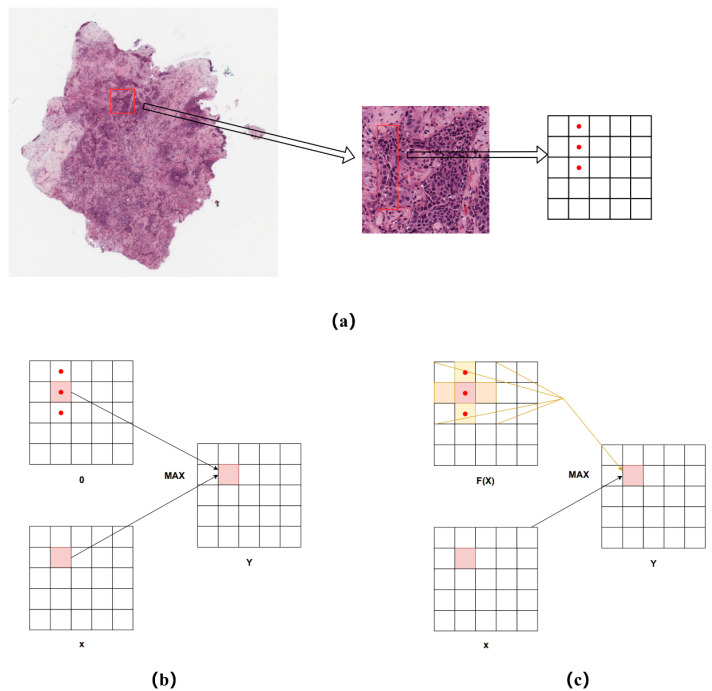
Visual activation function integrated into spatial information. (**a**) receptive field area of pathological image feature map of lung cancer; (**b**) ReLU with a condition zero; (**c**) CroReLU with a space parametric condition.

**Figure 4 cancers-14-05181-f004:**
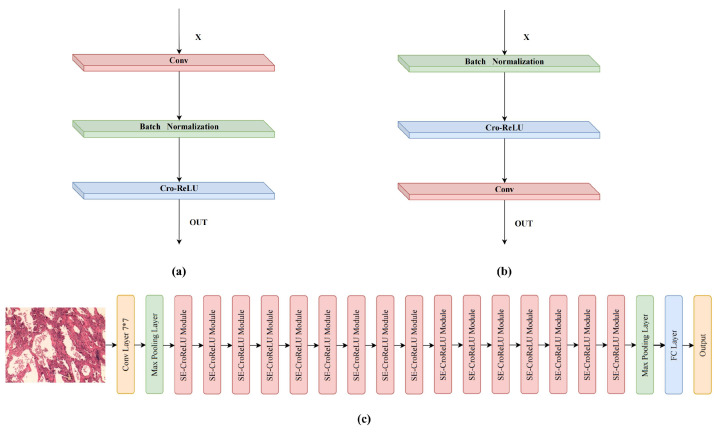
In combination with CroRelu’s convolutional network module: (**a**) Conv-ReLU:Conv-BN-AF; (**b**) CroReLU-Conv: BN-AF- Conv; (**c**) Overall architecture of SENet50_CroReLU.

**Figure 5 cancers-14-05181-f005:**
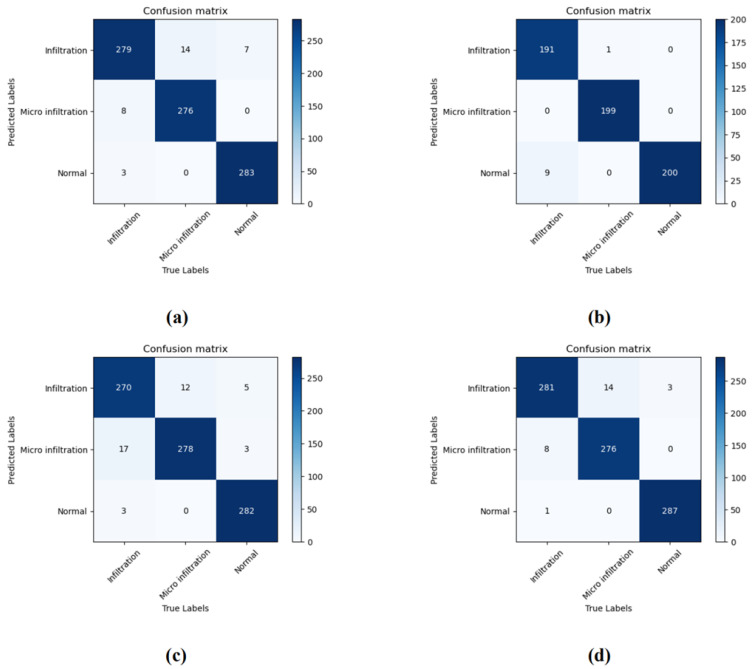
The confusion matrix obtained by the model on a private dataset. (**a**) SENet50; (**b**) SENet50_CroReLU; (**c**) MobileNet; (**d**) MobileNet_CroReLU.

**Figure 6 cancers-14-05181-f006:**
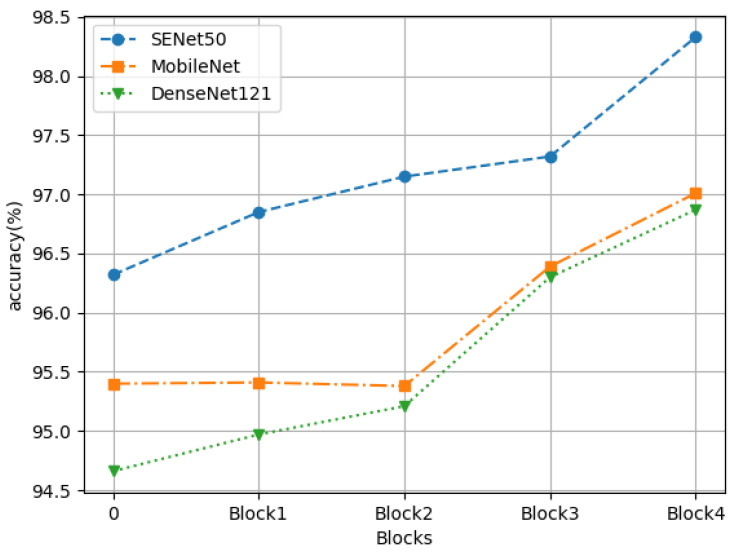
Impact of the number share of CroReLU on accuracy on three deep learning models.

**Figure 7 cancers-14-05181-f007:**
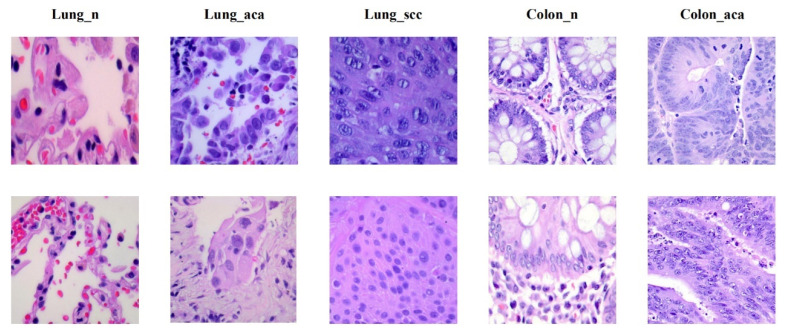
Dataset LC25000. From left to right: benign lung pathology image (Lung_n), lung adenocarcinoma pathology image (Lung_aca), lung squamous carcinoma pathology image (Lung_scc), benign colon pathology image (Colon_n) and colon adenocarcinoma pathology image (Colon_aca), and the image size is 768×768.

**Figure 8 cancers-14-05181-f008:**
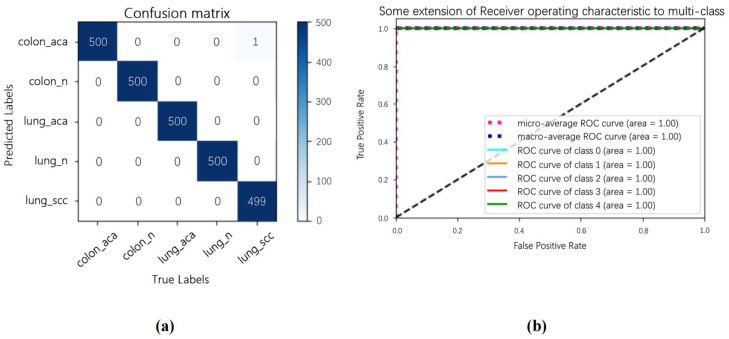
SENet50 _CroReLU classification results on LC25000 data showing (**a**) confusion matrix and (**b**) ROC curves.

**Table 1 cancers-14-05181-t001:** Comparison of the accuracy of ReLU and CroReLU for lung cancer disease screening using two types of neural network models on a private dataset.

Methods	Accuracy	Precision	Sensitivity
SENet50	96.32	96.51	96.33
SENet50_CroReLU	98.33	98.38	98.35
MobileNet	95.40	95.46	95.42
MobileNet_CroReLU	97.01	97.07	97.04

**Table 2 cancers-14-05181-t002:** Ablation studies applying different crossover window sizes to CroReLU on a private dataset.

Methods	Accuracy	Parameters	Test Time
SENet	96.32	**25.5 M**	**0.372**
SENet_3 × 3	**98.33**	26.1 M	0.393
SENet_5 × 5	97.26	26.5 M	0.478
SENet_7 × 7	97.09	27.2 M	0.505

**Table 3 cancers-14-05181-t003:** Data distribution of samples in the five categories of LC25000.

Image Type	Train	Test	Sum
Lung_n	4500	500	5000
Lung_aca	4500	500	5000
Lung_scc	4500	500	5000
Colon_n	4500	500	5000
Colon_aca	4500	500	5000

**Table 4 cancers-14-05181-t004:** Performance of the CroReLU method compared with existing works on the LC25000 dataset.

Authors	Accuracy(%)	Precision(%)	Sensitivity(%)	Remark
Masud M. et al. [29]	96.33	96.39	96.37	inter-class recognition
Mangal S. et al. [30]	Lung: 97.89 Colon: 96.61	-	-	intra-class recognition
B.K.Hatuwal et al. [31]	97.20	97.33	97.33	intra-class recognition
**Proposed**	99.96	99.87	99.86	inter-class recognition

Inter-class recognition (lung and colon): models can identify five subtypes of two cancers simultaneously. Intraclass recognition (lung or colon): models can only identify subtypes corresponding to a particular type of cancer.

## Data Availability

The public dataset is available in a publicly accessible repository. Please see [28]. The private dataset is not available because of privacy regulations.

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
