# Peer review of "CroReLU: Cross-Crossing Space-Based Visual Activation Function for Lung Cancer Pathology Image Recognition"

_cancers, 2022, doi:10.3390/cancers14215181_

Round 1

Reviewer 1 Report

This manuscript provides an interesting study of automatic detection of cancer infiltration level using deep learning methods on histological pathological images. The authors presented a machine learning approach for capturing model spatial information and  histological morphological features of lung cancer. The results show that the combined use of CroReLU and models had superior results than the models alone.I agree to recommend this article for publication with the following modifications:

1.     Use a consistent style for “CroReLU”. The reviewer saw at least two styles throughout the manuscript (e.g., CroReLU, Cro-ReLU).

2.     Could you please check your references carefully? All references must be complete before the acceptance of a manuscript.

3.     The senet should be briefly described by the authors in Neural network model.

4.     What is CroReLU's competitive advantage over the existing Vision Transformer model?

5.     The author claims that the extended experiment has proved that the model has strong generalization ability, so is it still necessary to perform data augmentation?

6.     The number of test sets should be 500 instead of 200.

7.      There are a few mistakes within the text. E.g. P11L357. A thorough read through for formatting, grammar etc is recommended. 

Author Response

Our modification suggestions are in the attached document.

Reviewer 2 Report

Page 4, lines 160-161: "In addition, patches with blurred, over-stained, and full-slide backgrounds were screened out through a manual process during the data pre-processing stage".

I believe the decision to exclude flawed samples from the analysis represents a bias in the selection of pathology images.

The authors aim to optimize time by reducing the workload for pathologists, but, if a human operator has to manually select the "good" slides to show to the AI tool, then I don't know how much time is actually gained. Human intervention is however necessary and cannot be eliminated. It would perhaps be more appropriate if the A.I. tool were able to select the good ones from a Patient's group of pathology images and then to perform the analysis.

I think this issue needs to be discussed and clafiried.

Author Response

Thank you for your comments concerning our manuscript.  We have studied comments carefully and have made correction which we hope meet with approval.  Our modification suggestions are in the attached document.
